# Cyromazine Effects the Reproduction of *Drosophila* by Decreasing the Number of Germ Cells in the Female Adult Ovary

**DOI:** 10.3390/insects13050414

**Published:** 2022-04-27

**Authors:** Muhammad Zaryab Khalid, Zhipeng Sun, Yaoyao Chen, Jing Zhang, Guohua Zhong

**Affiliations:** Key Laboratory of Natural Pesticide and Chemical Biology, Ministry of Education, South China Agricultural University, Guangzhou 510642, China; zaryabkhalid0003@hotmail.com (M.Z.K.); szp617488580@outlook.com (Z.S.); chenyy@stu.scau.edu.cn (Y.C.); zhangjing@scau.edu.cn (J.Z.)

**Keywords:** insecticide, toxicology, oogenesis, reproduction, germline stem cells, cystoblasts, ecdysone signaling, RT-qPCR, hormone titer, immunofluorescence staining

## Abstract

**Simple Summary:**

Cyromazine, an insect growth regulator, is used to control the Dipteran pest population. Previous findings observed that treatment with cyromazine increased the larval mortality, by interfering with the ecdysone signaling. In addition, the application of exogenous 20E significantly reduced the mortality caused by cyromazine. Many studies have also supported the role of ecdysone signaling in the maintenance of germline stem cells (GSCs), where mutations in ecdysone signaling-related genes significantly decreased the number of GSCs. However, to date, no study has reported the effect of cyromazine on the GSCs of *Drosophila melanogaster*. In the present study, we observed that cyromazine significantly reduced the number of both GSCs and cystoblasts (CBs) in the ovary of adult female. To further understand the effect of cyromazine on germ cells, we selected some key genes related to the ecdysone signaling pathway and evaluated their expression through RT-qPCR. Additionally, we measured the ecdysone titer from the cyromazine-treated ovaries. Our results indicated a significant decrease in the expression of ecdysone signaling-related genes and also in the ecdysone titer. These results further supported our findings that cyromazine reduced the number of germ cells by interfering with the ecdysone signaling pathway.

**Abstract:**

In the present study, we observed a 58% decrease in the fecundity of *Drosophila melanogaster*, after treatment with the cyromazine. To further elucidate the effects of cyromazine on reproduction, we counted the number of both germline stem cells (GSCs) and cystoblasts (CBs) in the ovary of a 3-day-old adult female. The results showed a significant decrease in the number of GSCs and CBs as compared to the control group. The mode of action of cyromazine is believed to be through the ecdysone signaling pathway. To further support this postulate, we observed the expression of key genes involved in the ecdysone signaling pathway and also determined the ecdysone titer from cyromazine-treated ovaries. Results indicated a significant decrease in the expression of ecdysone signaling-related genes as compared to the control group. Furthermore, the titer of the ecdysone hormone was also markedly reduced (90%) in cyromazine-treated adult ovaries, suggesting that ecdysone signaling was directly related to the decrease in the number of GSCs and CBs. However, further studies are required to understand the mechanism by which cyromazine affects the GSCs and CBs in female adult ovaries.

## 1. Introduction

Proper maintenance and differentiation of germline stem cells (GSCs) are very important for an organism’s reproduction [1]. The ovary of *Drosophila* is an excellent system to study GSCs in vivo as the system is tractable, GSCs loss is adequately observed, and the germ cells can be easily identified based on their position and available molecular markers [2]. The niche has been known to play crucial roles in GSCs identity maintenance in organisms, extending from *Drosophila* to mammals [3]. Further, the niche of *Drosophila*’s ovary is composed of terminal filament (TF) cells, cap cells, and escort cells. These somatic cells maintain the GSCs fate by producing many short-range signals [4,5]. Under the action of these signals, GSCs divide asymmetrically and produce cystoblasts (CBs). Subsequently, these CBs undergo mitotic division with incomplete cytokinesis four times and result in cyst formation. Only one cell among the cyst differentiates into an oocyte, while the remaining 15 cells differentiate as nurse cells (Figure 1). Somatic cells also send signals to the cyst at this stage that affect germ cell division [6]. Thus, the niche not only maintains GSCs but also controls the differentiation of germ cells. 

Cyromazine (N-cyclopropyl-1,3,5-triazine-2,4,6-triamine) is a triazine insect growth regulator used to control the insect pest population. It is a cyclopropyl derivative of melamine which is believed to affect the development of larvae and pupae by disrupting cuticle turnover during ecdysis [7,8]. *Stomoxys calcitrans* and *Lucilia cuprina* are among the important livestock pests which are effectively controlled by cyromazine. Further, it is also used against *Musca domestica* [9]. Treatment of *Drosophila*’s larvae with cyromazine caused early eclosion of adults, and it is believed that cyromazine affects the larval development by affecting the ecdysone signaling pathway [10]. 

To date, very little understanding is available on the role of the ecdysone hormone in post-developmental processes such as ovary development. Recent studies have confirmed that the ovary itself produces ecdysteroids which are necessary for oogenesis, niche formation, GSCs maintenance, and cyst differentiation [11,12]. First, 20E triggers the heterodimeric nuclear hormone receptor which is encoded by *EcR* and *usp* genes. Later, this heterodimer controls the expression of ecdysone-responsive genes, by binding to a specific promoter [13,14]. As the ecdysone signaling pathway is very complex and involves many genes, we therefore randomly selected some 20E responsive genes which significantly regulate germ cells in the germarium of *D. melanogaster*. The mutations in these selected genes (*EcR*, *usp*, *E75B*, *E78*, *sad*, *spok*, *kr-h1*, *nvd*, *tpr2*, *vkg*, *Hrb27C*, and *CycE*) markedly decreased the number of germline stem cells [15]. The downstream genes of the ecdysone signaling pathway, specifically *spook* and *phantom*, are also found necessary for proper ovary development [16]. Besides the target of ecdysone signaling, *Tpr2* is also an important regulator of fecundity [17,18]; while the loss of *CycE* from the GSCs and CBs resulted in a block in the cell cycle [19]. Furthermore, *Hrb27c* is found to be ecdysone responsive and also a vital regulator of GSCs maintenance. The mutation of *Hrb27c* in the germarium of *D. melanogaster* resulted in a 63% loss of GSCs [15,20]. Additionally, it has been reported that the 20E also regulate *Kr-h1* which supports reproduction by promoting oocyte maturation and ovary development [21,22]. Further, ecdysone signaling assures proper egg production by acting as a developmental checkpoint over mid-oogenesis [23]. 

In the present study, we observed that the continuous selection of cyromazine significantly affected the germ cells at the adult stage. To the best of our knowledge, this is the first report of an insect growth regulator affecting the germ cells in insects. In addition, we observed the expression of ecdysone signaling-related genes and also determined the ecdysone titer to better understand the effect of cyromazine on GSCs and CBs.

## 2. Materials and Methods

### 2.1. Drosophila Strain and Breeding Conditions

All the experiments were conducted by using the *Drosophila melanogaster* y w strain. Both larvae and adults were reared on a standard cornmeal agar medium at a constant temperature of 25 °C and humidity of 75% with a 12:12 h light/dark cycle. To prevent bacterial and fungal contamination, we mixed 10% acetyl acid in a freshly prepared diet. 

### 2.2. Insecticide Treatment

Technical-grade cyromazine (active ingredient) was purchased from Guangzhou Qixiang Biotechnology Co., Ltd. Briefly, a 1 mg/mL stock solution of cyromazine was prepared in distilled water to make the serial dilutions. Subsequently, the freshly prepared diet was mixed with a specified volume of cyromazine, and *D. melanogaster* was continuously selected from mid-3rd instar larvae until 3-day-old adults. Further, mid-3rd instar larvae were allowed to feed on a diet containing a low concentration of cyromazine (0.3 PPM). However, virgin female adults were allowed to feed on a diet containing a higher concentration of cyromazine (50 PPM) for 3 days. Distilled water mixed in the diet was used against the control.

### 2.3. Immunohistochemistry and Microscopy

For performing immunohistochemistry, 3-day-old female adult ovaries were dissected in phosphate buffer saline (PBS) under a stereoscopic microscope. The ovaries were then fixed in 4% paraformaldehyde (PFA) for 50 min. Later, the samples were washed 3 times with PBST (0.1% Triton X-100 in PBS), each time for 10 min. Blocking was performed in 5% normal goat serum (NGS) for 1 h, and then the samples were incubated with primary antibodies at 4 °C overnight. The following day, PBST was used to wash the samples 3 times and then blocking was again performed in 5% NGS. Later, the secondary antibodies were added and the samples were incubated for 2 h. To stain the DNA, we further used Hoechst (1:5000; Cell Signaling Technology, Danvers, MA, USA). The mounting of ovaries was performed in 90% glycerol. A total of 300 germaria were counted for a single representative experiment of three replicates (100 germaria/replicate).

Among the primary antibodies, we used: mouse anti-α-Spectrin (3A9, AB_528473, 1:100; Developmental Studies Hybridoma Band [DSHB], Iowa City, IA, USA) and rabbit anti-pMad (13820T, 1:400; Cell Signaling Technology), while mouse 488 (1:1000) and rabbit cy3 (1:1000) were used as secondary antibodies. For taking the images, a Nikon A1 plus confocal microscope was used (Nikon, Tokyo, Japan).

### 2.4. Counting the Number of GSCs and CBs in the Ovary of 3-Day Old Female Flies

The expression of pMad is highly specific for GSCs [24]. Therefore, we used rabbit anti-pMad (1:400; Cell Signaling Technology) as a primary antibody to label the GSCs in the germarium of the ovariole. Further, we differentiated CBs with the differentiating cysts based on the presence of round spectrosomes. For this purpose, we used mouse anti-α-Spectrin (3A9, 1:100; Developmental Studies Hybridoma Band [DSHB]) as a primary antibody (Figure 2). 

### 2.5. Effect of Cyromazine on the Expression of Ecdysone Signaling Related Key Genes 

Following treatment with cyromazine, quantitative real-time PCR (qRT-PCR) was performed to determine the expression of 12 ecdysone signaling-related genes (*EcR*, *usp*, *E75B*, *E78*, *sad*, *spok*, *kr-h1*, *nvd*, *tpr2*, *vkg*, *Hrb27C*, and *CycE*). The adult ovaries (N = 20) were dissected to extract total RNA (1 µg) and reverse-transcribed into cDNA according to the manufacturer’s instructions (Takara, Beijing China). Later, SsoFast Eva Green Supermix (Bio-Rad, Hercules, CA, USA) was used to perform qRT-PCR (Bio-Rad iQ2 optical system). A total reaction mixture of 25 μL was used, while rp49 was used as an internal control to normalize the expression of ecdysone signaling-related genes by the 2^−ΔΔ^CT method. The conditions used were: 95 °C for 2 min, 40 cycles each of 95 °C for 5 s, 60 °C for 10 s, and melting curve from 65–95 °C [25,26]. Further, 3 biological replicates were used and each biological replicate was repeated 3 times with 20 ovaries per replicate. Primer Premier 5 (software) was used to design gene-specific primers and are provided (Appendix A).

### 2.6. Ecdysone Titer Measurement 

The ecdysone titer was measured from both the larval and adult ovaries. To measure the ecdysone titer from adult ovaries, 20 ovaries were dissected for each replication; while in the case of larvae, 300 larval ovaries were dissected for each replication. Furthermore, three replicates were used for each experiment. Briefly, the ovaries were dissected in PBS from the treated and control groups and immediately stored at −20 °C until needed. Later, the ovaries were homogenized and centrifuged supernatant was used to measure the ecdysone hormone titer by using an insect ecdysone enzyme-linked immune sorbent assay (ELISA) kit (Shanghai MLBIO Biotechnology Co., Ltd., Shanghai, China) according to the manufacturer’s instructions and following previous studies [27]. The absorbance was measured at 450 nm on a 96-well microplate photometer (Multiskan™ FC, Thermo Fisher, Waltham, MA, USA). 

### 2.7. Effect of Cyromazine on the Ovary Morphology and Fecundity

The effect of cyromazine on the morphology of the ovary was observed by measuring the length and diameter of the ovary after treatment with cyromazine; while the effect on the fecundity was observed by counting the number of laid eggs by the 3-day-old adult. Briefly, 20 virgin females of the y w strain were kept with the males and allowed to mate for 24 h. Later, the male flies were removed, and 20 mated female flies (3-day-old) were allowed to lay eggs (24 h) on the sugar-grape-based medium in the Petri dish, for each replication. Further, 3 replications were used. The eggs were counted under a Leica microscope. 

### 2.8. Statistical Analysis

Microsoft Excel and Graphpad Prism 9.0 were used for data collection and statistical analysis. For *p*-values, Student’s *t*-test was used, with: ** p* < 0.05; ** *p* < 0.01; *** *p* <0.001, indicating statistically significant differences. 

## 3. Results

### 3.1. Cyromazine Effects the GSCs and CBs in the 3-Day Old Female Adult Ovary

*D. melanogaster* was continuously selected with the cyromazine from mid-3rd instar larvae until 3-day-old adults. Later, the ovaries of the 3-day-old female adults were dissected and immunofluorescence staining was performed. The germarium of the *Drosophila* was stained with both anti-α-Spectrin (3A9) and anti-pMad. pMad is specifically expressed in GSCs. Therefore, the number of GSCs was counted based on the expression of pMad, while spectrosomes with a single spherical shape were present on the GSCs and CBs. Therefore, the number of CBs was counted based on the presence of such single dot-like spectrosomes. Our results showed that the number of GSCs significantly decreased (1.73 ± 0.34) as compared to the control group (2.37 ± 0.21) (Figure 3A). Furthermore, a 23% decrease in the number of CBs was also observed in the germarium of *Drosophila* treated with cyromazine (Figure 3B). 

### 3.2. Expression of Ecdysone Signaling Related Key Genes

Ecdysteroids are crucial for the proper maintenance and proliferation of GSCs [15,28]. Therefore, we evaluated the mRNA expression of some key ecdysone signaling-related genes. Our results showed that the expression of most genes was downregulated, as compared to the control. Among 12 ecdysone signaling-related genes, the expression of only *E75B* and *kr-h1* was upregulated; while the expression of the remaining selected genes was downregulated. The largest decrease was observed against *tpr2* followed by *vkg*, *sad*, *CycE*, *Hrb27C*, *spok*, *nvd*, *EcR*, *E78*, and *usp*, respectively (Figure 4). Thus, these results further supported that cyromazine might affect the GSCs and CBs in the female adult ovary by affecting the expression of ecdysone signaling-related genes. 

### 3.3. Cyromazine Effects the Ecdysone Titer in Both the Larval and Adult Ovaries

The mechanism of action of cyromazine is hypothesized to be related to the ecdysone signaling pathway [10]. Therefore, to support the effect of cyromazine on GSCs and CBs, we measured the ecdysone titer from both the late 3rd instar larval and 3-day-old female adult ovaries. To our knowledge, no study has reported the ecdysone titer from the larval ovaries of *D. melanogaster*. Our results indicated a significant decrease in the ecdysone hormone level in both the larval and adult ovaries as compared to the control group (Figure 5). Among the treated larval ovaries, about a 50% decrease in the ecdysone titer was observed as compared to the control group (Figure 5A); while in the adult ovaries, the ecdysone titer decreased from 235 ± 39 ug/mL (control group) to 25.01 ± 18 ug/mL (treated group) (Figure 5B). These results indicated that cyromazine significantly affected the ecdysone titer in the ovaries of *D. melanogaster*, and further supported our findings that cyromazine affected the GSCs and CBs by affecting the ecdysone signaling. 

### 3.4. Effect on the Ovary Morphology and Fecundity

We further observed the effect of cyromazine on the ovarian morphology of the 3-day-old adult female. A significant decrease in the size of the ovary was observed after treatment with cyromazine (Figure 6). 

To evaluate the effect of cyromazine on the fecundity of *D. melanogaster*, 20 female adults were allowed to mate and lay eggs on the sugar-grape-based medium in the Petri dish for 24 h. Later, the number of eggs was counted. The results indicated a 58% decrease in the fecundity of cyromazine-treated female flies as compared to the control group (Figure 7A).

## 4. Discussion

In *D. melanogaster*, ecdysteroids control larval development [29,30]. Whereas, in adults, these hormones are found to be essential for proper oogenesis [31,32]. Furthermore, different developmental processes such as vitellogenesis, egg production, and egg development are strictly regulated under the presence of ecdysteroid signaling [33,34]. The ecdysone hormone titer has also been determined from virgin adult ovaries, indicating their importance in virgin females [35,36]. In addition, the ecdysone hormone is vital for the maintenance of GSCs [37,38]. 

Both reproduction and metamorphosis, in insects, are negatively regulated by cyromazine [39]. It is believed that cyromazine affects insect development by interfering with the ecdysone signaling pathway. In *Drosophila*, the application of cyromazine increased larval mortality, while the exogenous treatment of 20E decreased larval mortality. Further, 20E treatment together with cyromazine increased the early eclosion of adult flies [30]. In the present study, we observed a 27% decrease in the number of GSCs after treatment with cyromazine. Whereas a 23% decrease in the number of CBs was observed as compared to the control group. 

The treatment of insecticides, during different developmental stages of insects, has a significant effect on the mRNA expression level [40]. A recent study reported that insecticides significantly affected the mRNA expression of *EcR* and *usp* [41], while in *Drosophila* larvae the treatment of cyromazine reduced ecdysone signaling [10]. However, the response of ecdysone signaling to insecticides is poorly understood. To the best of our knowledge, no study has explored the effects of cyromazine on the germ cells of *D. melanogaster*. Therefore, in the present study, we tried to investigate the effect of cyromazine on the germ cells by evaluating the expression of key ecdysone signaling-related genes, and also by determining the ecdysone hormone titer. The expression of ecdysone genes (*shd*, *spo*, *sro*, *nvd*, *nobo*, and *phm*) has been reported to increase significantly just after 18 h of female adult eclosion, indicating the regulation of ecdysone genes in virgin females. Further, *nvd* plays a crucial part in the first process of the ecdysteroid biosynthesis pathway. When *nvd* was silenced by RNAi, the number of GSCs significantly decreased as compared to the control group [36,42]. Further, both *EcR* and *usp* are broadly expressed in the germline stem cells. In addition, *EcR* alone is vital for proper oogenesis. Mutation in *EcR* resulted in the development of abnormal egg chambers [43]. Likewise, the reduction in the expression of *EcR*, *usp*, and *E75* significantly inhibited cyst development [37]. However, when the expression of *vkg*, *tpr2*, *Hr39*, *trx*, and *CycE* was reduced using mutant flies, 66%, 44%, 35%, 40%, and 50% GSCs loss was observed, respectively [15]. Our results are in agreement with previous findings where the expression of ecdysone-responsive genes was downregulated, implying that the treatment of cyromazine decreased the mRNA expression of ecdysone-responsive genes which in turn was related to the decreased number of germ cells in the ovary. 

In addition to GSCs maintenance and niche formation, ecdysone signaling is also required for proper germline differentiation. Disturbance in the ecdysone signaling notably hindered germline differentiation [44,45]. Similarly, in the present study, we observed that the expression of most ecdysone signaling-related genes was downregulated in cyromazine-treated insects. 

Furthermore, we found that the ecdysone titer was significantly decreased in cyromazine-treated larval and adult ovaries. In larval ovaries, about a 50% decrease in the ecdysone titer was observed. However, around a 90% decrease in the ecdysone titer was seen against the adult ovaries as compared to the control group. Previous studies have reported that the ecdysone hormone directly controls GSCs maintenance and resurgence [14,46] supporting our results that cyromazine reduced GSCs and CBs by interfering with ecdysone signaling. 

In addition, we observed a 58% decrease in the fecundity of the cyromazine-treated female adults as compared to the control group. The proper expression of ecdysone signaling is necessary for normal oogenesis [35,47]. Such results strengthen our findings and further support the role of ecdysone signaling in the reproduction of *D. melanogaster*. 

Furthermore, the treatment of cyromazine might also interfere with other signaling pathways such as the juvenile hormone (JH) pathway. In insects, the interaction between ecdysone and JH is very complex, through which they support normal development, growth, and reproduction; while JH has been known to negatively regulate the ecdysone signaling pathway [48]. Furthermore, it is reported that JH repressed the ecdysone by *Kr-h1* [49,50]. We also observed such interesting results in our present study, where the expressions of most of the ecdysone signaling-related selected genes were downregulated. However, the expression of *Kr-h1* was significantly increased. This might be due to the effect of cyromazine where JH decreased the ecdysone titer through *kr-h1*. In summary, the present results provided a more detailed understanding of ecdysone signaling-related genes in terms of insecticide-responsive genes, after treatment with cyromazine. However, further studies are required to deeply understand if cyromazine affects the germ cells of *D. melanogaster* by interfering with other signaling pathways such as the JH signaling pathway. 

## 5. Conclusions

The current study provides crucial information on the effect of an insect growth regulator on the germ cells of the female adult ovary. To the best of our knowledge, it is the first report presenting the effect of cyromazine on the GSCs and CBs of the female adult ovary. To further support our results that cyromazine affected the germ cells by interfering with ecdysone signaling, we determined the ecdysone hormone titer from the treated ovaries. The results showed about a 90% decrease in the ecdysone titer from the adult ovaries. Furthermore, the expression of most of the ecdysone signaling-related genes was also downregulated as compared to the control group. These results support our findings that cyromazine might affect the germ cells by interfering with ecdysone signaling. However, further studies are required to understand the mechanism through which cyromazine affects the germ cells in the female adult ovary. 

## Figures and Tables

**Figure 1 insects-13-00414-f001:**
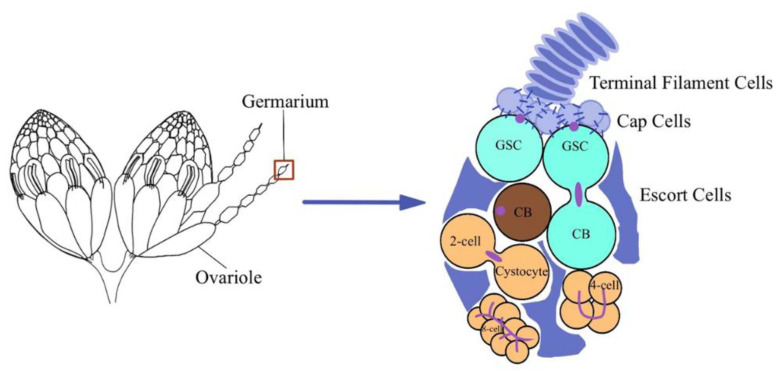
Different types of somatic and germ cells in the germarium of *Drosophila.* The ovary of *Drosophila* consists of 16–20 ovarioles, and the anterior region of each ovariole is known as germarium. Germarium houses different types of somatic cells (terminal filaments, cap cells, and escort cells). These somatic cells are vital for the maintenance of germ cells (germline stem cells, cystoblasts). In addition, GSCs are anchored with the cap cells which help them to retain their stem cell character. Further, the GSCs are directly connected with the escort cells which are also necessary for their proper maintenance.

**Figure 2 insects-13-00414-f002:**
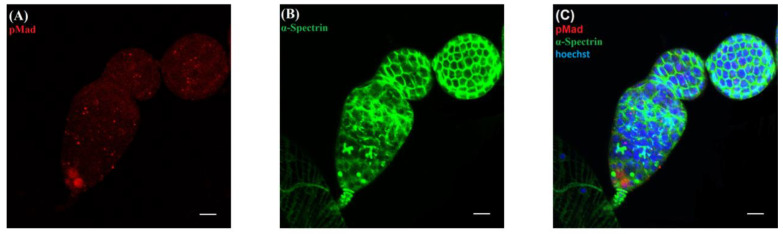
(**A**) For staining the GSCs, pMad antibody was used. (**B**) For staining single dots containing spectrosomes and fusomes, we used α-Spectrin. (**C**) CBs were differentiated from the GSCs and cyst based on the presence of single spectrosome-containing dots. Only GSCs and CBs have single dots containing spectrosomes. To stain DNA, Hoechst was used. Scale bar, 10 μm.

**Figure 3 insects-13-00414-f003:**
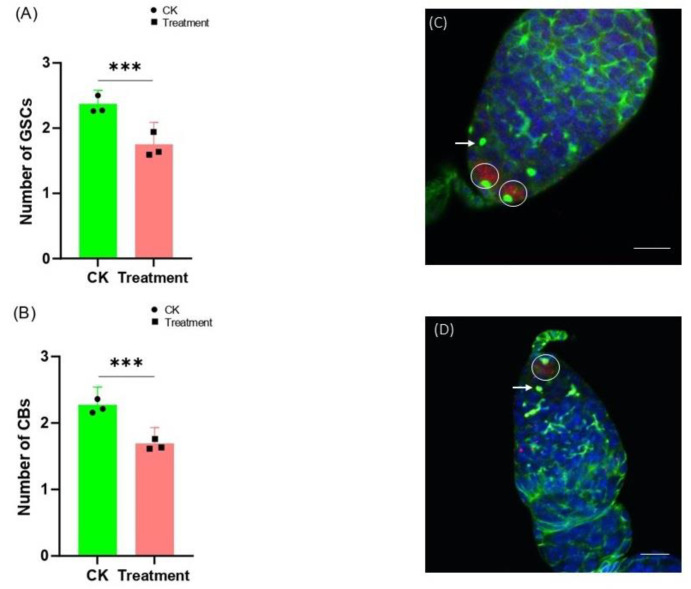
Effect of cyromazine on the germ cells of female adult ovary. *D. melanogaster* was continuously selected with cyromazine from the mid-3rd instar larvae until 3-day-old adults. Later, the number of germ cells were counted from 100 germaria per biological replicate (N = 300). (**A**) Results showed a 27% decrease in the number of GSCs compared to the control group. (**B**) A 23% decrease in the number of CBs was observed. (**C**,**D**) The germarium of both control and treated groups, respectively, were stained with pMad, α-Spectrin, and Hoechst. Further, the GSCs are indicated by white circles and the CBs are indicated by white arrows. Scale bar, 10 μm. The *p*-value of Student’s *t*-test is *** *p* < 0.001.

**Figure 4 insects-13-00414-f004:**
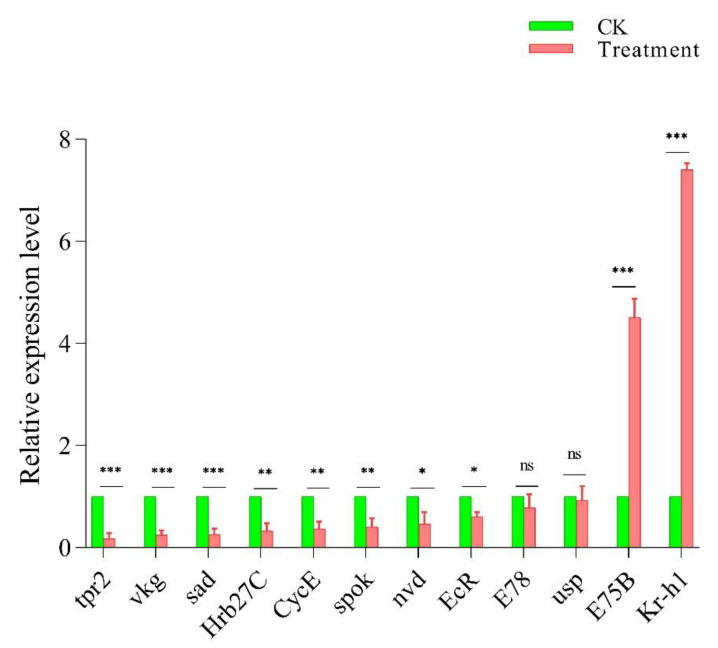
Expression of ecdysone signaling-related genes against cyromazine treatment. Error bars indicate 95% confidence intervals (CI). The *p*-value of Student’s *t*-test is: * *p* < 0.05; ** *p* < 0.01; *** *p* < 0.001; ns not significant.

**Figure 5 insects-13-00414-f005:**
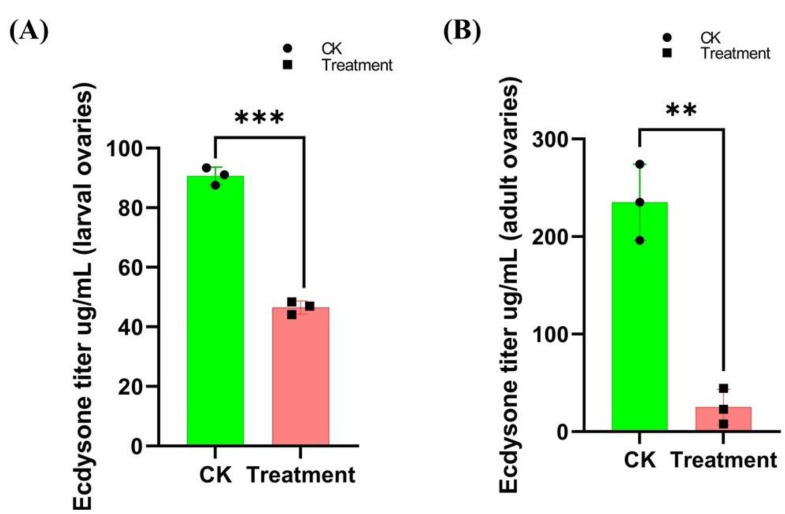
Effect of cyromazine on the ecdysone titer of larval and adult ovaries. (**A**) Ecdysone titer from the larval ovaries. Due to the small size of the larval ovaries, a total of 300 larval ovaries were dissected for each replication (N = 900, total number of larval ovaries dissected for each experiment). (**B**) For determining ecdysone titer from the adult ovaries, 20 ovaries were dissected for each replication (N = 60, total number of adult ovaries dissected for each experiment). The *p*-value of Student’s *t*-test is: ** *p* < 0.01; *** *p* < 0.001.

**Figure 6 insects-13-00414-f006:**
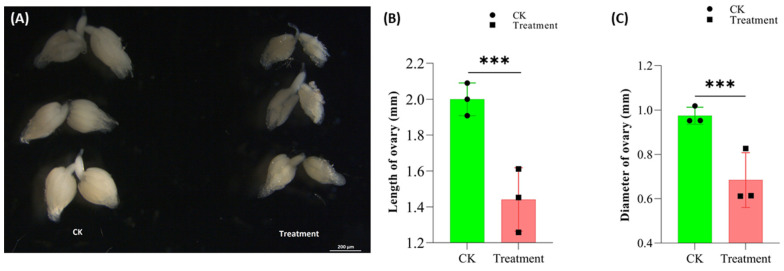
Effect of cyromazine on the ovary size of *Drosophila*. (**A**) Cyromazine significantly decreased the ovary size as compared to the control group. (**B**) The length of the ovary decreased from 1.97 ± 0.21 mm to 1.45 mm ± 0.22. A total of 6 ovaries were dissected for each replication (N = 18). (**C**) The diameter of the ovary decreased from 0.97 ± 0.13 mm to 0.69 ± 0.17 mm in the cyromazine-treated group compared to the control group (N = 18, as 6 ovaries were dissected for each replication). The *p*-value of Student’s *t*-test is: *** *p* < 0.001.

**Figure 7 insects-13-00414-f007:**
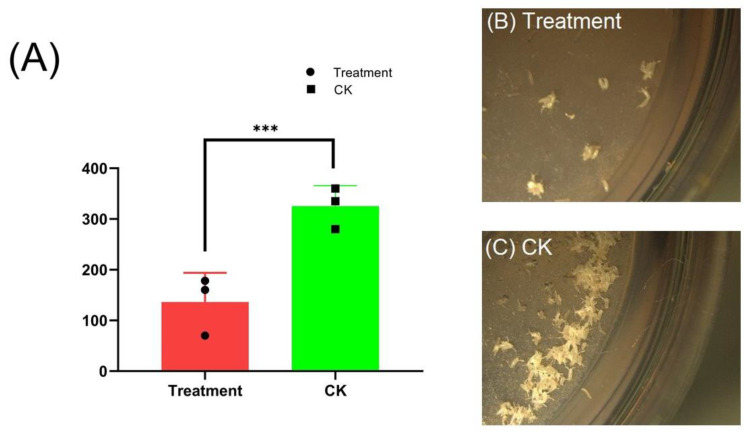
Effect of cyromazine on fecundity of *Drosophila melanogaster*. (**A**) Twenty mated female flies were used for each replication (N = 60). The number of laid eggs decreased from 325 ± 40.9 (CK) to 136 ± 57.8 (Treatment). (**B**,**C**) Significant decrease in the fecundity of the 3-day-old female adult was observed, as compared to the control group. The *p*-value of Student’s *t*-test is: *** *p* < 0.001.

## Data Availability

The data presented in this study are available in article or Appendix A.

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
