# Peer review of "Cyromazine Effects the Reproduction of Drosophila by Decreasing the Number of Germ Cells in the Female Adult Ovary"

_insects, 2022, doi:10.3390/insects13050414_

Round 1

Reviewer 1 Report

Kalid et al assess the effects of an insect pesticide, cyromazine on the physiology of wildtype Drosophila melanogaster, focusing on the ecdysteroid production and the female germline. They expose flies to the chemical at low doses during the L3 period or adults at much higher doses and find that the germline stem cells are affected as appears to be ecdysone levels.

Overall, there could be a number of improvements made to the quality and rigor of the work, but there are findings that are likely of interest to those who study pesticides and their effects, provided the below points are addressed. Clarifications in particular regarding replication and the execution of experiments are most critical, followed by the consideration and inclusion of experimental limitation/other possible explanations.

Major points to address:

  • Line 97 – specify how long L3 were exposed to chemical. Ie. was it until adulthood, or dissection? If the latter, when was this performed? At precisely what stage? This is important for understanding the consequences of its addition.
  • For all figures, please plot individual datapoints in figures so that replicates can be seen individually, and include N in the legend.
  • The effect on ecd-related genes is interesting but there are several reasons that it might not relate to GSCs and CBs – for instance, if the pesticide impaired nutritional signals from other parts of the body which caused slower maturation of oocytes.
  • Figure 5 and Section 3.3. It is impressive to measure L3 Ecd titres, however I am concerned about the interpretation of the outcome for both L3 and adults. What efforts were taken to normalise ecdysone levels to tissue volume? If growth is perturbed during L3 (and perhaps the timing of metamorphosis), as would be expected upon feeding cyromazine, how can one meaningfully compare treated vs untreated?
  • Similarly with adults, reproduction is sensitive to many factors, as I point out above for Ecd-response genes, couldn’t this huge difference be simply caused by independent factors that impinge upon ovary maturation and productivity? I am interested in the results here but remain sceptical that these effects aren’t due to the large physiological consequences of pesticide exposure.

Minor points to address:

  • May be journal-software-related but Figure 1 and others are a too small relative to the page.
  • Cyromazine introduction makes it sound like a gene product. To clarify this, I suggest the authors introduce it as the type/class of pesticide and provide some information regarding its source or production, and also its hypothesised mechanism of action since this is relevant to the present work.
  • Line 65 needs citation or connect it with the following sentence if ref 8 is the source of both pieces of information.
  • Line 68 use ‘To date’, rather than ‘Till date’.
  • Line 75 genes spook and phantom should be italicised.
  • Final sentence in introduction is very broad and seems to be off the topic of the present manuscript – which aims to provide a better understanding of the effects of pesticides on insect physiology.
  • Line 108, specify Hoechst concentration
  • Line 112 – specify animal species (e.g. instead of ‘ms’ write ‘mouse’), and company for secondaries.
  • Figure 2 – lots of white space. Enlarge images to reduce wasted space and make viewing easier.
  • Section 2.5 – specify numbers of ovaries per replicate and the number of biological replicates.
  • Section 2.6 – explain how (system used) the tissue was kept from degrading during the dissection of 300 L3 ovaries.
  • Section 2.7 – clarify whether 20 mated females were kept in separate enclosures, or together (if together, how may replicates were performed?), and more importantly whether males were also exposed to the chemical or not.
  • Figure 3. CK is not defined – what is it and how does it relate to the treatment? Ie. is it the solvent used for the cyromazine?
  • Panels C, D are not well labelled – arrows indicating GSCs and CBs would be helpful since here it is not clear what is being counted. Were these quantifications blinded?
  • The exposure period and duration should be included in the results section and figure legends.
  • Section 3.2 - Line 176 First sentence is not a sentence.
  • Section 3.3 - Line 192-3. First sentence is not a sentence.
  • There are many issues with the referencing formatting (including, but not limited to, Journal names are missing).

Author Response

Dear reviewer, please see the attachment. We thank you for your valuable suggestions in improving the manuscript.

Reviewer 2 Report

Cyromazine effects the reproduction of Drosophila by decreasing the number of germ cells in the female adult ovary– Kong et al.

Major revision

In this paper the authors investigated the effect of the insecticide cyromazine in the Drosophila melanogaster reproduction. In particular, after cyromazine treatment by feeding, they observed a reduction in germ cells and, therefore, an altered number of eggs laid. Although the manuscript is well presented, I have major concerns about the data analysis performed in some experiments and the choice of some experiments as well.  

Line 86.

The experiments were performed on D. melanogaster y w strain. Was there a reason the authors chose this strain over a wild type strain (eg. Canton-s)?

Line 92.

Since cyromazine is a cuticle synthesis inhibitor, has the treatment induced any different in the larvae development and adult formation?

Line 99.

How many insects the authors used for this experiment?

Line 110-113.

Please provide the Ab codes.

Figure 2 and figure 3.

Please add white arrows indicating the GSCs and CBs cells in the IHC pictures.

Line 126.

The author should explain (here or in the introduction) why they chose these specific gene target in the qPCR experiments and what roles they exert in the D. melanogaster reproduction.

In this section, the authors did not present adequate information: what was the reaction set up? What was the amount of RNA reverse-transcribed? How many replicates were performed? How many ovaries were dissected for each replicate? Furthermore, the authors used the Livak method to normalize the data on the housekeeping gene, but why they showed the data as relative expression level? And why was present only one control?

Line 137.

Was each replicate composed by 20 ovaries?

I have other two major comments:

  • In this paper the authors investigated the ecdysone pathway. Why did the authors not explore all the Halloween genes but only few of these? Interesting, the authors investigated the effect of cyromazine treatment on Kr-h1, the ecdysone response genes inhibitor. But Kr-h1 is the last actor of the juvenile hormone pathway. Why the authors did not consider to analyse these elements also (Met, Tai for example)? Furthermore, the authors should discuss the unusual strong upregulation of Kr-h1 after cyromazine treatment. In fact, if the cyromazine induced a reduced ecdysone production, why the inhibitor is overexpressed?
  • Since ecdysone signaling might be implicated in Vg synthesis, why the authors did not explore the effect of cyromazine in the protein production, total protein uptake in ovaries, egg hatching ratio?

The discussion is very concise and I thought the authors could discuss more in deep the receptor expression profile after cyromazine treatment. Furthermore, the authors should consider to discuss the implications that the cyromazine treatment might be have on other reproduction-related pathways (eg Juvenile hormone pathway).

Author Response

(The authors gave the same response as above.)

Round 2

Reviewer 2 Report

The authors have addressed all my concerns.

Author Response

Esteemed reviewer, the manuscript has been revised thoroughly according to your kind suggestions. We are very thankful to you for your valuable suggestions in improving the manuscript. 

This manuscript is a resubmission of an earlier submission. The following is a list of the peer review reports and author responses from that submission.